# Furnishing a Recreational Forest—Findings from the Hallerwald Case Study

Renate Cervinka [1,2,†], Markus Schwab Spletzer [1,2,3,†] and Daniela Haluza [2,*]

[1] Institute for Corporate Governance, Research and Innovation and National Higher Education Cooperation, University College for Agrarian and Environmental Pedagogy, 1130 Vienna, Austria; renate.cervinka@haup.ac.at (R.C.); markus.schwab2@gmx.at (M.S.S.)
[2] Department of Environmental Health, Center for Public Health, Medical University of Vienna, 1090 Vienna, Austria
[3] Pro Mente Research, 1050 Vienna, Austria
[*] Correspondence: daniela.haluza@meduniwien.ac.at
[†] These authors contributed equally to this work.

**Abstract:** While the beneficial effects of forests on health and well-being are broadly investigated, little is known on the restorative effects of forest infrastructure. Thus, this study assessed the perceptions of installing furniture in a recreational forest in forest visitors. We surveyed 220 volunteers attending guided walks before ($n$ = 99) and after ($n$ = 121) furnishing the Hallerwald. The questionnaire assessed restorative qualities of four places in the forest before and after furnishing, and changes in visitors' self-perceptions pre and post visiting the forest for 2.5 h. Further, visitors evaluated the furniture and the visit. The four sites in the forest under study benefited differently from furnishing. We found mixed outcomes with respect to the restorative qualities of places by furnishing, and a similar improvement of human restoration pre- and post-walk, irrespective of furnishing, but received mainly positive ratings for the installed furniture. The participants expected positive effects of visiting the forest to last one to two days. Our findings suggest that furnishing the forest made this forest a unique place for pedagogy, health interventions, and tourism. We concluded that furnishing, designed to fit the characteristics of a specific place, can support health and well-being in restorative forests and should be recognized by sustainable forest management.

**Keywords:** citizen science; design; furniture; green care; psychological restoration; green public health; regional development; soft forest furnishing; sustainable forest management

## 1. Introduction

In addition to the economic and ecological function of forests, the social, health-promoting, and therapeutic function of forests has gained increasing attention in research in recent years [1–4]. A considerable body of scientific literature shows the restorative power of forests [2–10]. Outcomes such as improved health and well-being [4,9,11,12], decreased stress, improved human restoration [3,13], improved reflection [14], brightened mood [15–18], increased vitality [19], stronger connectedness with nature and the forest, as well as improved mindfulness [20–22] are empirically proven. Recent systematic reviews have underpinned the effectiveness of forest therapy programs [2,3]. In the past few years, the interest in visiting recreational forests and the establishment of healing forests has been increasing [10]. This is accompanied by efforts to furnish such forests. Therefore, it is important to investigate the influence of furnishing on the recreational function of forests and to make recommendations on design.

Numerous studies have explored the physical characteristics of forests and their relevance to forest design and management. In particular, certain forest characteristics and specific locations within the forest have been found to correlate with psychological restoration. Exemplarily, international evidence has been gathered on differences in wild or tended

forests [16], different ages of forests, management style, location [6,7], and various stand types [23]. Perceived sensory dimensions [13], sounds and personal characteristics [19], and different seasons [8,24,25] were investigated with respect to beneficial effects on health and well-being. Solitude and forest settings with light were identified as positive factors for recovery of humans suffering from mental exhaustion [26]. Signposts are commonly used to stimulate experiences such as immersion in forests. A Finnish study investigating well-being trails in different countries showed the positive effects of psychological interventions [15]. Results showed changes in restorative experience and mood. Visitors reported high satisfaction with the trails and a willingness to recommend the trail to friends. They were also satisfied by the number of signposts provided. Further studies investigated the influence of educational boards and guides in relation to forest visits [27–29]. They also reported recreational effects.

Forest infrastructure is any kind of element that makes a forest accessible and operational to visitors [10,30,31]. This includes parking lots outside the forest, comfortable and safe paths, benches, and signposts that help to attract users and make their visit a pleasant experience. While beneficial effects of forests on health and well-being have been comprehensively investigated, little is known on the restorative effects of furnishing forests. Initial data suggest that recreational facilities in forests will not necessarily increase the restorative potential for visitors. Infrastructure in forests can even impair the positive effects of forests if conflicting with the needs and intended preferences of users.

A survey conducted in Germany investigating recreational forest infrastructure in 600 forest visitors concluded that recreational facilities (e.g., benches) are not necessary at all to the forest users, as their primary motive was taking in the atmosphere of the forest [31]. Participants of telephone surveys conducted in Switzerland reported no particular need for recreational infrastructure, and that heavily used infrastructure by other user groups impaired their own restoration [30]. A study conducted in Norway found that participants preferred images of forest scenes that showed little human influence or infrastructure [32]. On the other hand, providing furniture that fits the specific characteristics of a place might even increase the restorative properties of a place in the forest [33]. Summing up, sensory experiences typical in a forest have turned out to be the main reason for visiting the forest and infrastructure may disturb this experience. We suspect that providing furniture fitting the specific characteristics of a place might even increase the restorative properties of a place in the forest.

Although there is some scientific evidence of the lastingness of forest therapy programs [34], little is known on the lastingness of restorative outcomes of simple forest visits. Exposure–response relationships are generally under-researched. The first beneficial effects on psychological parameters arise after a few minutes in green environments [35]. However, for good health and well-being at least 120 min of direct contact to nature are recommended [36]. Therefore, further high-quality investigations on the lastingness of forest visits are needed.

In order to continue the findings of current research on health and well-being effects of forests and on furniture, we aimed at further contributing to the body of knowledge on the beneficial effects of forests as well as on the specific case of furniture in forests, with respect to the public health of local people and health tourists. Our current study focused on furnishing, i.e., on equipping a restorative forest with recreational features. In a prior study, we investigated the restorative qualities of four different places in the forest, as well as well-being parameters, connectedness with nature, and mindfulness pre and post walking in the forest for approximately 2.5 h [21]. The study followed a citizen science approach, involving the local community in study planning, and data collection. All places under study reached high scores in perceived restorativeness. We found that positive mind states improved through visiting the forest while negative effects and perceived stress decreased. The biggest change was the improvement of the perceived restorative outcome in visitors.

The present study utilizes data from the prior study, which reflects the conditions prior to furnishing. The design of the furnishing was inspired by theories from environ-

mental psychology [37–39]. The elements should be functional, safe, and inviting; facilitate aesthetic and affective responses; and support the given restorative potential of the place. Concurrently, the furniture should neither diminish nor disturb the naturalness of the location. Sustainable forest management emphasizes, among other things, the psychosocial function of the forest and the preservation of biodiversity [10]. In line with this concept, we assumed that recreational facilities should support restoration and not spoil the forest characteristics. So, this study aimed at investigating changes before and after furnishing with respect to restorative qualities of four different places in the forest, and the self-perceptions of visitors pre and post visiting these places in the forest. Additionally, we were interested in evaluations of the forest visit with respect to aspects of satisfaction with the furniture and perceived lastingness of participants' restoration after visiting the furnished forest.

## 2. Materials and Methods

### 2.1. Study Design and Procedure

The study was conducted in a community forest located in the Austrian county of Upper Austria, called Hallerwald. This forest covers an area of 270,000 m$^2$ and is used for wood production, education, and recreation. The vegetation corresponds to a mixed forest with areas of spruce monoculture. Forest owners chose four places to be studied. Sites had to represent different stand types and show various characteristics of the forest. Further, they had to be accessible by different user groups, such as people in different phases of life or those with health and mobility restrictions. Therefore, the four places under study were all located near a well-kept forest path and were easily accessible for visitors. These places represented different green and blue spaces, stand type, openness, and biodiversity. The purpose of the local authorities was to enhance the potential of the Hallerwald forest by installing furnishings that not only improve the quality of recreation, but also emphasize the uniqueness of the forest. As such, the aim was to maintain the forest as a recreational space for the local population while also making it accessible for health tourism and sustainable education. In a prior study, we reported on the restorative qualities of unfurnished places in the forest [21].

The furnishings used were designed to increase the forest's attractiveness to visitors, who would be delighted by the specially designed places within the forest. The design of the furnishings was inspired by theories from environmental psychology, and great care was taken to ensure that they were tailored to the specific characteristics of the forest [37–39]. The furniture should be functional, safe, and inviting, facilitate aesthetic and affective responses, and support the given restorative potential of the place. Overall, the furniture should support soft fascination. The idea of soft furnishing forests contrasts with the implementation of adrenaline-releasing infrastructure, such as mountain bike trails or high ropes courses, which we would classify as provoking "hard fascination".

The furnishing of the four places under study occurred between 2018 and 2019. Locals made the furniture by hand, so each piece of furniture was unique. Figure 1 presents images of the places before and after equipping with furniture.

The first place (Figure 1A) represents a mixed blue and green space. A small creek flows in a ditch in the terrain and washes around spherical stones covered with moss. The place is called Mossy Stones. Mossy Stones was equipped with a solid wooden platform. The platform located above the ditch in the terrain was designed to protect the sensitive geological structure, but also to meet the safety requirements of visitors. They should experience the place with all their senses, but not climb down to the creek.

The second place (Figure 1B) represents a glade in the forest covered with ferns and horsetails. It is a place of great biodiversity surrounded by a mixed vegetation of young and old trees. The place is called Fern Glade. To preserve the uniqueness of the place, a sign was put up asking visitors neither to enter the place nor to damage the sensitive vegetation. In the vicinity, stable wooden constructions with nets for sitting or lying in enabled the sensual experience and relaxation of the visitors.

**Before Furnishing**          **After Furnishing**

(**A**) Mossy Stones

(**B**) Fern Glade

(**C**) Outlook

(**D**) Forest Glade          (**D'**) Willow Arbor

**Figure 1.** Four sites under study before and after furnishing.

The third place (Figure 1C) opens up a view of the landscape into a valley characterized by farmland and high mountains in the distance. It is just outside the forest. The place is called Outlook. A small country road leads past it. The furniture consists of three wooden benches and an information board nearby.

The fourth location in the prior study (Figure 1D), named Forest Glade, represented a clearing in the forest. The ground is uneven and covered with brown spruce needles. Wild blueberry bushes and moss form the vegetation next to tall spruce trees. This place typically for this forest should remain unchanged. Very close by, there was a level place for designing a willow dome. This place offered itself for re-design after a previous dying back and clearing of ash trees. The former place (Figure 1D) now is called Willow Arbor (Figure 1D'). A delicate metal framework supports the wicker vegetation, which will stabilize itself when mature. In the front third of this construction, a mighty tree stump is located. This tree stump reinforces the impression of a green spiritual place.

### 2.2. Data Collection and Study Sample

We collected questionnaire data during summer 2018 [21] and summer 2019. Recruitment was achieved using advertisements in newspapers, on the homepages of cooperating partners' websites, and by involving local clubs and unions. Inclusion criteria for voluntary participation were: (1) age older than 18 years of age, (2) ability to walk a forest trail for at least 2.5 h, and (3) giving written informed consent prior to participation in the study. Additionally, (4) immunization against tick-born encephalitis was required, as it is a public health threat in Austria, and we did not want to put participants in danger of getting sick. In total, 222 (before furnishing, $n = 99$; after furnishing, $n = 121$) volunteers participated in the study. The characteristics of the two samples are shown in Table 1. In 2019, participants took part in the guided walks in groups of two to a maximum of eleven.

**Table 1.** Characteristics of the study sample.

| | Before Furnishing (n = 99) ° | | After Furnishing (n = 121) | | Total (n = 220) | | p |
|---|---|---|---|---|---|---|---|
| | **M** | **SD** | **M** | **SD** | **M** | **SD** | |
| Duration of the visit (minutes) | 164.37 | 19.65 | 164.14 | 12.96 | 164.25 | 16.28 | 0.948 [a] |
| Age (years) | 43.15 | 17.11 | 43.30 | 15.38 | 43.23 | 16.18 | 0.130 [a] |
| | *n* | % | *n* | % | *n* | % | |
| Sex (female) | 63 | 63.6 | 89 | 73.6 | 152 | 69.1 | <0.05 [b] |
| Visits to Hallerwald | | | | | | | |
| First time | 23 | 23 | 46 | 38 | 69 | 31.4 | <0.05 [b] |
| Occasionally | 49 | 50 | 52 | 43 | 101 | 46.0 | |
| More than once a year | 19 | 19 | 12 | 9.9 | 31 | 14.1 | <0.05 [b] |
| More than once a month | 7 | 7 | 8 | 6.6 | 15 | 6.8 | |
| Several times a week | 1 | 1 | 2 | 1.7 | 3 | 1.4 | |
| Every day | 0 | 0 | 1 | 0.8 | 1 | 0.5 | |

Note: [a] U-test, [b] z-test. ° Mean (M), SD, *n*, and % before furnishing have already been reported in [21] and are shown here for convenience reasons.

Both studies followed the same procedure: researchers trained guides in briefing participants on the procedure of the tour, and how to manage the questionnaires. Prior to the guided walk, lasting for approximately 2.5 h, and connecting the four different places in the forest, participants received a bottle of water and additional information. Data collection took place at each place as well as pre- and post-walk. At each of the respective places, participants were instructed to explore the place with all their senses for ten minutes. The participants received the identical instruction at each place. The aim was to support the participants in perceiving the natural conditions prevailing at the location as comprehensively as possible.

In contrast to earlier research [15], the aim of the instruction was not to focus on the participants' personal state of mind in relation to mood or the experience of stress. It also did not aim to create mental images, find metaphors, or reinforce possible well-being effects. The instruction was intended to refer exclusively to the conditions on site and was

formulated neutrally. Participants were asked not to speak to each other while experiencing the places but were free to contact the guide at any time if they had any questions. Guides reminded the participants to fill out the questionnaires after 10 min, so that checking the time does not disturb participants while experiencing the places. There were no rules of conduct on the paths between places besides those that apply in forests in general. Groups rated the places in rotated order (figures in parentheses report the absolute frequencies before setup, with place D before furnishing corresponding to D' after furnishing): 50 (34) participants visited the places in order ABCD, 32 (22) in order BCDA, and 39 (34) in order DCBA. It was assured that the walking distance between the places was roughly the same. The order of the visits to the places could not be fully rotated, as the terrain did not allow for this.

*2.3. Measures*

Data on age and sex were collected as sociodemographic data. Further, the frequency of visits was assessed, with one item covering six categories ranging from visiting the forest "for the first time" (1) to "visit the forest every day" (6). We used Cronbach's alpha ($\alpha$) to calculate internal consistency of scales. The qualities of the places were Perceived Restorativeness Potential (PRP) and the potential to Widen One's Mind (WOM) [21]. Participants responded on an 11-point answering format from 0 "not true at all" to 10 "completely true" for all the respective scales. We used a short version of the Perceived Restorativeness Scale (PRS) [40,41] to assess the PRP, which consisted of four items of the subscale Being away and four of the subscale Fascination. Internal consistency ($\alpha$) was 0.92 for place (Figure 1A) Mossy Stones, 0.86 for place (Figure 1B) Fern Glade, 0.90 for place (Figure 1C) Outlook, and 0.90 for place (Figure 1D') Willow Arbor. We assessed the potential to Widen One's Mind by applying the WOM scale, comprising 6 items [21]. Internal consistency $\alpha$ was 0.93 for place (Figure 1A) Mossy Stones, 0.91 for place (Figure 1B) Fern Glade, 0.93 for place (Figure 1C) Outlook, and 0.95 for place (Figure 1D') Willow Arbor.

We assessed positive and negative affect, perceived stress, perceived restoration, connectedness to nature, and connectedness to the forest as well as mindfulness pre- and post-walk. A short version of the Positive Negative Affect Schedule (PANAS) [42] measured affect; $\alpha$ was 0.85 for positive affect pre-visit, and 0.89 post-visit. For negative affect, pre-visit $\alpha$ was 0.84, and 0.87 post-visit. A single item assessed perceived stress, asking participants how stressed they felt at the moment. Participants responded on an 11-point response format from 0 "not true at all" to 10 "completely true". Three items of the Restoration Outcome Scale (ROS) [43] measured perceived restoration with an 11-point response format from 0 "not true at all" to 10 "completely true"; pre-visit $\alpha = 0.88$, post-visit $\alpha = 0.89$. The Inclusion of Nature in the Self scale (INS) assessed connectedness to nature [44]. Respondents marked one of seven graphics representing their relationship to nature best. We assessed Connectedness with the Hallerwald using a modified version of the INS, replacing only the term "Nature" with the term "Hallerwald". A 4-item version of the Freiburg Mindfulness Inventory [45] assessed mindfulness. Internal consistency was $\alpha = 0.72$ pre-visit and 0.71 post-visit.

After completing the walk in the furnished forest, participants were asked to estimate in days, hours, and minutes how long the restorative effects of the walk would last. To assess how the participants compared the restorative effect of the forest walk in contrast to other recreational activities, we used a single item asking them: "How do you rate the effect of visiting the forest in comparison with other recreational activities (e.g., jogging, walking, swimming, watching TV, go to a movie, play videogames)?" We used a 5-point response format ranging from "much smaller" to "much stronger". Regarding the evaluation of the experience, participants responded to three single items with an 11-point response format from 0 "don't agree" to 10 "completely agree". They were asked about their willingness to re-visit the forest, to recommend a visit to friends, and how satisfied they were with the forest visit before and after furnishing. Participants were asked to evaluate the furniture with respect to its effect on the appearance of the place, its perceived safety of use, the extent

to which it provided a sense of refuge and facilitated social interactions, and if it made the stay at the place a special experience. All items applied an 11-point response format.

To investigate if the furnishing of the places impaired the beneficial effects of the place, we created a scale consisting of 4 items. "The furnishing elements reduce the experience of nature", "The furnishing elements affect the appearance of the place", "The furnishing elements spoil the impression of the place", and "The furnishing elements reduce the value of the place"; α was 0.89 for place Figure 1A, 0.80 for place Figure 1B, 0.81 for place Figure 1C, and 0.85 for place Figure 1D'. Further, participants rated with single items how safe the furniture was to use, and how much the furnishing elements supported the sense of refuge. Two items assessed if the furnishing elements facilitated social activities and interactions at the place, "The furnishing elements encourage activities with other people", and "The furnishing elements stimulate conversation". The Spearman–Brown coefficient was 0.87 for place Figure 1A, 0.89 for place Figure 1B, 0.85 for place Figure 1C, and 0.82 for place Figure 1D'. Additionally, participants rated if the furnishing made the place something special.

*2.4. Statistical Analysis*

We used SPSS Version 27 (IBM Corp., Armonk, NY, USA) for all analyses. Differences in the characteristics of the samples before and after furnishing were analyzed using Fischer's exact test and Mann–Whitney U-tests when required. We conducted KS tests and visual inspection of Q-Q-plots to assess normality. To investigate differences in the rating of the restorative qualities of the places between, before, and after furnishing, we specified two identical general estimating equations (GEE) [46,47] with PRS and WOM as outcome variables. Because the sample showed differences in gender and frequency of visits between before and after furnishing, these variables were entered into the model as controls. We used an autoregressive working correlation matrix (AR1) accounting for within-subject correlations. Significant interaction between place and wave would indicate an effect of the furnishing of the forest. For post hoc comparisons of the places between the waves, we followed Fisher's least significant differences procedure (LSD).

Differences between self-perceptions before and after the walk were analyzed using independent *t*-tests, comparing the mean pre- and post-walk differences before and after furnishing. To investigate differences regarding ratings of the furnishing, we conducted RM-ANOVAs for each variable of interest. Information on satisfaction and lastingness of the forest experience was investigated using descriptive statistics and *t*-test or U-tests or RM-ANOVA when appropriate. Analysis of missing data was conducted for the whole dataset on an item level. No pattern in missing values (1.74%) was detected by Little's MCAR test ($\chi^2$ (679) = 604.19, *p* = 0.982) [48]. However, some clusters were detected visually. In these cases, a whole page was missing, so we assumed that participants had inadvertently skipped a page of the questionnaire. In the assessment before furnishing, two faulty copies led to this pattern of missing values [21].

**3. Results**

*3.1. Restoratives Qualities of the Places before and after Furnishing*

For the restorative qualities of the four places in the forest, participants rated the qualities of places regarding PRP and WOM. Table 2 shows ratings of PRP and WOM for the four places in the forest before and after furnishing.

We conducted GEEs for PRP and WOM to investigate the effects of equipping the places with furniture. Before furnishing, Fern Glade followed by Mossy Stones showed higher values in PRP compared to the Forest Glade and the Outlook. With respect to WOM, the Fern Glade scored significantly higher than all other places. After furnishing, Fern Glade showed the highest scores for PRP and WOM among all places under study. In both years, Outlook scored lowest for PRP and for WOM. The next paragraph reports on differences in the scores of the places before and after furnishing for PRP and WOM.

**Table 2.** Qualities of the places in the forest and the forest on average.

| | **Before furnishing** | | | | | | | | | |
| | **(A)** Mossy Stones | | **(B)** Fern Glade | | **(C)** Outlook | | **(D)** Forest Glade | | **(A–D)** Total Forest | |
| | **M** | **SD** | **M** | **SD** | **M** | **SD** | **M** | **SD** | **M** | **SD** |
| PRP | 7.34 * C, D | 1.42 | 7.70 * C, D | 1.41 | 6.15 A, B | 1.69 | 6.65 A, B, C | 1.74 | 6.96 | 1.11 |
| WOM | 7.67 B, C | 1.40 | 8.05 * A, C, D | 1.38 | 6.69 A, B, D | 1.94 | 7.54 B, C | 1.55 | 7.49 | 1.16 |
| | **After furnishing** | | | | | | | | | |
| | **(A)** Mossy Stones | | **(B)** Fern Glade | | **(C)** Outlook | | **(D′)** Willow Arbor | | **(A–D′)** Total Forest | |
| | **M** | **SD** | **M** | **SD** | **M** | **SD** | **M** | **SD** | **M** | **SD** |
| PRP | 6.86 * B, C | 2.01 | 8.37 * A, C, D | 1.21 | 6.21 A, B, D | 1.77 | 6.91 B, C | 1.76 | 7.09 | 1.69 |
| WOM | 7.31 B, D | 1.87 | 8.79 * A, C, D | 1.10 | 6.61 A, B, D | 2.08 | 7.43 B, D | 1.81 | 7.54 | 1.72 |

Note: M, mean, SD, standard deviation; PRP, Perceived Restorative Potential; WOM, Widen One's Mind. Subscript letters indicate significant ($p < 0.05$) post hoc comparisons within year (Fischer's LSD). * Indicate significant differences before and after furnishing. Means and SD before furnishing have already been reported in [21] Table 1 and are shown here for convenience reasons.

The KS test found no significant deviation from normality before furnishing. However, the KS test showed significant deviation from normality for PRP of Mossy Stones (D (120) = 0.098, $p = 0.007$), and for PRP of Fern Glade after furnishing (D (120) = 0.106, $p = 0.002$). An examination of respective Q-Q-plots suggested moderate skewness. We found a significant interaction between place and wave (Wald Chi-Squared (7) = 309.974, $p < 0.001$). A post hoc analysis revealed that Fern Glade scored significantly higher on PRP after furnishing ($p < 0.040$). Mossy Stones scored significantly lower ($p < 0.001$) after furnishing. There were no significant differences in the other places with respect to PRP.

The KS test showed significant deviation from normality of WOM for Fern Glade (D (99) = 0.096, $p = 0.026$) before furnishing. The KS test showed significant deviation from normality of WOM for Fern Glade (D (121) = 0.137, $p < 0.001$) and for Willow Arbor (D (121) = 0.108, $p = 0.001$) after furnishing. An examination of respective Q-Q-plots suggested moderate skewness. We found a significant interaction between the place and wave (Wald Chi-Squared (7) = 217.078, $p < 0.001$). A post hoc analysis revealed that Fern Glade scored significantly higher on WOM after furnishing ($p < 0.001$), while there was no significant difference in all other places.

*3.2. Comparison of Changes in Self-Perceptions before and after Furnishing*

Our study aimed to evaluate changes in several psychological parameters of forest visitors, including positive and negative affect, perceived stress, perceived restoration, connectedness with nature and the forest, and mindfulness, both before and after visiting the forest. To assess the impact of furnishings on these parameters, we compared pre- and post-visit scores and controlled for gender and visit frequency using a multiple regression model. As the significance of the interaction term was not changed, we considered the influence of gender and visit frequency negligible in this analysis and reported *t*-test results.

The changes in visitors' self-perception before and after visiting the forest were similar before and after the furnishing. The differences in self-perception did not reach statistical significance. The effect sizes of the differences were small or negligible. Table 3 shows the changes in scores before furnishing and after furnishing. Perceived restoration showed the highest increase in both study years before and after furnishing. Connectedness to the forest showed the second highest increase, followed by reduction in perceived stress and positive affect. For all other variables, the change before and after furnishing was less than 10%. So, furnishing was not reflected in the self-perception data.

**Table 3.** Comparison of changes in participants' self-perceptions.

| | Before Furnishing ° | | After Furnishing | | | |
|---|---|---|---|---|---|---|
| | **M** | **SD** | **M** | **SD** | ***p*** | **Effect Size** |
| Positive affect | −10.39 | 17.83 | −11.01 | 16.88 | 0.793 | 0.04 a |
| Negative affect | 8.95 | 19.40 | 7.97 | 14.67 | 0.678 | 0.06 b |
| Perceived stress | 15.11 | 25.73 | 15.78 | 20.50 | 0.834 | 0.03 a |
| Perceived restoration | −24.68 | 21.98 | −25.29 | 21.20 | 0.840 | 0.03 a |
| Connectedness to nature | −8.60 | 15.48 | −8.48 | 19.99 | 0.962 | 0.01 a |
| Connectedness to forest | −21.01 | 25.86 | −24.78 | 22.98 | 0.266 | 0.16 a |
| Mindfulness | −6.81 | 15.37 | −3.88 | 13.34 | 0.138 | 0.20 a |

Note: All scores are POMP-transformed; *p* values from *t*-tests; effect sizes reported as Hedge's g av (a) and Hedge's g rm (b). ° M, mean, and SD, standard deviation, before furnishing have already been reported in [21] Table 2, and are shown here for convenience reasons.

### 3.3. Assessment of the Furnishing

Our study focused on evaluating the furniture installed in the furnished forest. To assess its effectiveness, visitors were asked to evaluate the furniture in terms of how well it complemented the characteristics of each individual space, as well as its overall impact on the quality of the forest. Additionally, visitors were asked to provide feedback regarding the safety of the furniture (Table 4). We explored the impact of the furniture on the quality of the forest by examining different aspects of quality, ultimately answering the question of how much the furniture enhanced the overall quality of the forest. With respect to impairing the quality of the places, the furnishing of the places was perceived differently (Greenhouse–Geisser F (2.766, 326.432) = 15.573, *p* < 0.001). Post-hoc tests showed Mossy Stones as being most disturbed by furnishing compared to all other places. Fern Glade was impacted the least by furnishing.

**Table 4.** Ratings of the furniture at the places.

| | (A) Mossy Stones | | (B) Fern Glade | | (C) Outlook | | (D') Willow Arbor | | (A–D') Furnishing Average | |
|---|---|---|---|---|---|---|---|---|---|---|
| | **M** | **SD** | **M** | **SD** | **M** | **SD** | **M** | **SD** | **M** | **SD** |
| Impairing quality | 3.00 B, C, D | 2.52 | 1.38 A, C, D | 1.84 | 1.98 A, B | 2.09 | 2.27 A, B | 2.19 | 2.18 | 1.49 |
| Sense of refuge | 4.65 B, D | 2.86 | 8.63 A, C, D | 1.75 | 5.11 B, D | 2.83 | 7.06 B, C, D | 2.51 | 6.36 | 1.65 |
| Stimulation of social interaction | 5.73 C | 2.60 | 5.91 C | 2.94 | 7.75 A, B, D | 2.11 | 6.36 C | 2.49 | 6.44 | 1.72 |
| Special experience | 5.62 B, C, D | 2.89 | 8.97 A, C, D | 1.46 | 7.02 A, B, D | 2.49 | 7.93 A, B, C | 2.21 | 7.38 | 1.66 |

Note: M, mean, SD, standard deviation; subscript letters indicate significant (*p* < 0.05) post hoc comparisons (Fisher's LSD). Low ratings represent low impairment of the quality of the place.

The furnishing of the places was perceived differently with respect to providing a sense of refuge (Greenhouse–Geisser F (2.767, 315.392) = 79.272, *p* < 0.001). Fern Glade, followed by Willow Arbor, showed significantly higher scores compared to Outlook and Mossy Stones. In addition, the furnishing of the places was perceived differently regarding providing stimulation of social interaction (Greenhouse–Geisser F (2.685, 308.752) = 20.430, *p* < 0.001). Outlook scored statistically significantly best with respect to stimulation of social activities compared with all other places.

Participants' ratings of furnishing significantly differed with respect to special experience (Greenhouse–Geisser F (2.679, 308.050) = 66.892, *p* < 0.001). The ratings for all locations were statistically different, with Fern Glade furniture rated best, followed by Willow Arbor, Outlook, and Mossy Stones furniture. With the exception of social stimulation, Fern Glade furnishings consistently scored highest. Furniture at all locations received more than nine

points in terms of their perceived safety. Overall, users perceived the newly constructed furniture as safe.

### 3.4. Evaluation of the Forest Visit

We further evaluated the forest visit, before and after furnishing with respect to touristic aspects. Regarding willingness to revisit and recommend to friends, no difference between the unfurnished and furnished forest was found in terms of willingness to revisit (M1: 8.71, SE1: 0.20; M2: 8.46, SE2: 0.21; t(209) = −0.455, *p* = 0.650), and recommending a visit to friends (M1: 8.62, SE1: 0.19; M2: 8.72, SE2: 0.16; t(210) = 0.829, *p* = 0.408).

Participants rated the furnished-forest visit in comparison to other leisure-time activities as follows: 28% rated it as much more restorative, 60% as more restorative, 7% as equally restorative, and one percent rated it as lower restorative. No participant rated the forest visit as much lower restorative compared to other leisure-time activities.

Regarding subjective estimation of lastingness, participants stated that they expected the beneficial effect of the furnished forest to last about two days (hours: M = 45.42, SD = 48.41, BCa 95% CI [36.64, 55.42]). However, visual inspection of the distribution of the estimates showed positive skewness: 30 persons (28.6%) estimated that the effect would last about half a day, 20 persons (19.0%) estimated that the effect would last up to one day, 31 persons (29.5%) estimated that the effect would last up to two days, 14 persons (13.3%) estimated that the effect would last up to three days, and 10 persons (10.0%) estimated that the effect would last more than three days.

### 4. Discussion

This study investigated effects of the furnishing of a recreational forest in two consecutive years, before furnishing and after furnishing. For this purpose, we examined four selected places in the Hallerwald with regard to their recreational effect, asked study participants about aspects of self-perception before and after visiting the forest, and also requested their assessment on the furnishing. Questions on the evaluation of the 2.5-h forest visit rounded off the survey. A previous study has demonstrated the restorative properties of the unfurnished Hallerwald in terms of the restoration, vitality and mental foresight, as well as on the mood, restoration, connectedness with nature, and mindfulness of visitors [21].

Furnishing this forest was intended to further enhance proven benefits. The design of the furniture and placing was inspired by theories from environmental psychology with respect to function, aesthetics, and restorative potential. Further, the furniture should be inviting and be used intuitively. The furnishing was designed by local people in a way to preserve, or even increase, the characteristics of places in the forest by fitting the furniture carefully to the places' properties. We use the term "soft forest furnishing" to describe this effort, highlighting the importance of the fit between the characteristics of the place and the shape of the furniture.

First, we investigated the change in the restorative qualities of places before and after furnishing. Prior studies on furnishing forests reported mixed results. On the one hand, furnishing was not perceived as necessary, or even counteracted restoration in forests [30,31]. On the other hand, furniture could even enhance the benefits of the forest by enabling sensory experiences [33]. Results from this study support the assumption from prior research of mixed results concerning furnishing.

Our results showed the potential of furnishing to influence the restorative qualities of the places in the forest with respect to Perceived Restorative Potential and the potential to Widen Ones' Mind. However, only one place, Fern Glade, profited significantly from furnishing. This place, however, had also reached high scores without furnishing. The original high value of restorative qualities at Fern Glade was improved further by furnishing. We consider this might be due to a perfect fit between the installed furniture and the place characteristics. The special design of the wooden construction, combining the function of a

chair and a hammock could support sensual experience of nature, as well as concentrating on the self, relaxation and on reloading ones' batteries.

In contrast, the mixed blue and green space called Mossy Stones significantly lost Perceived Restorative Potential. The potential to Widen One's Mind showed the same trend, but the change did not reach statistical significance. The platform near the water element was designed not only to protect the sensitive geological structure, but also to meet safety standards for users. These demands resulted in a quite stable structure, providing a more technical instead of naturally inviting appearance. We suspect that the platform could be perceived as a barrier between nature and visitors, instead of offering a good overview of the place and allowing for an experience of nature in its full range. After furnishing, the platform invited neither leisurely rest nor the touching of stones or water. These considerations are in line with findings in walkers who preferred locations where they could interact with natural elements [33].

Two other places were robust against furnishing with respect to significant changes in PRP and WOM scores. The newly built Willow Arbor scored slightly higher compared to the Forest Glade but did not meet the expectations with respect to significant improvements in psychological restoration. This could be attributed to two aspects. First, the Willow Arbor would also be experienced as monoculture; second, the metal framework was not completely overgrown in 2019. The vegetation would have needed more time to mature in order to give the look and feel of a place providing refuge. The Outlook just outside the forest showed the lowest scores before furnishing. Furnishing increased neither the Perceived Restorative Potential, nor the potential to Widen Ones' Mind. This is in line with our expectations that a place next to a forest, and near a small country road furnished with benches, seems to be suitable for social interaction, but might not be the best place to increase cognitive restoration or mind wandering.

Second, we investigated changes in self-perception. A stay of 2.5 h in the unfurnished forest resulted in a significant increase in positive affect, perceived restoration, connectedness to nature and the forest, and mindfulness. Negative affect and perceived stress decreased significantly. All parameters of self-perception changed, as expected, before and after visiting the unfurnished forest as part of a guided tour stimulating sensual experience. This also applied to participation in a similar walk through the furnished forest. Therefore, the furnishing had no significant influence on the change in parameters of self-perception. As the restorative qualities of the places did not benefit from furnishing on average, furnishing the forest might be of minor influence on individuals' mood and connectedness compared to the 2.5 h walk through the forest. These results support the idea that furnishing the forest might not be necessary for harvesting the beneficial effects of a forest walk [30,31].

Third, to complete our research on furnishing we will discuss the direct evaluation of the furniture at the places. Overall, users did not rate the restorativeness of the places as significantly affected by the furniture. However, in detail we found statistically different ratings regarding the direct assessment of the furniture. The direct assessments support the findings and discussions presented above. Again, the soft furniture of the Fern Glade turned out to impair the quality of the place less, provide the highest sense of refuge, and make the place a special experience. The Mossy Stone's facility was rated as having the highest reduction in quality compared to the other three sites. We consider this result as not surprising given the design and placement of the platform. However, it should also be noted that the average rating of the furniture at Mossy Stones was below the center of the disturbance rating. With respect to social activities, the furnishing at the Outlook was rated as most facilitating. Installed benches and the information board matched this place. We consider a social place in the vicinity of a forest might be well placed, as the effects of social activities do not interfere with forest serenity and forest sounds.

The survey covered the qualities of the places, self-perceptions of visitors, and evaluation of the furniture, as well as evaluation of the visit. This multifaceted approach allowed for a holistic assessment of the furniture in the forest. It made it possible to look at the



interaction between the furniture and the characteristics of the site. For the assessment of such furniture, we have learned that only surveys covering different aspects lead to valid results. In practice, however, retrospective evaluations are common from different perspectives. In such a case, it should be taken into account that even small disturbances by furniture can result in reduced restorative qualities of places. Further research could identify the restorative potential of soft forest furniture; for instance, by using an experimental design. We recommend studies that apply new technologies. Virtual environments can stimulate sensory impressions and so trace in detail the impact of design on the restoration of forest visitors. This may help to continue studies on restorative experience in forests and stand-type [6,7,23].

Notably, visitors of the Hallerwald showed a high willingness to visit the forest again, and a high commitment to recommend a visit to friends, both before and after furnishing. We identified neither positive nor negative changes in visitors' willingness to re-visit or recommend with respect to furnishing. We reported above that the furniture had no impact on visitors' self-perceptions. These findings were mirrored in visitors' willingness to revisit or recommend. With respect to other leisure time activities, participants rated the forest walk as more restorative or as much more restorative. The effects of forest walks might even be more pronounced in reality, taking affective forecasting into account. A study reported that the beneficial effects of walking in green space were underestimated compared to walking activities in built environments [18].

The lastingness of the effect of the forest visit appears as a further indicator to evaluate the forest visit. Therefore, we investigated subjective estimations on how long the beneficial effects of the forest walk would last. A study in children suggests that compared to other green spaces such as parks, forests had a more sustained effect on concentration [49]. A study on a forest therapy program found that relaxing effects lasted for 3–5 days [34]. We are not aware of studies providing evidence on the lastingness of beneficial effects of forests in healthy adults. We suppose that the effects of a dedicated forest therapy program might be more intense compared to a guided 2.5 h-walk in the forest. This is reflected in our findings, as most participants estimated the beneficial effects to last less than two days. However, the estimates differed strongly between participants, which might point to the existence of subgroups that systematically differ regarding the sustained effects of forest visits. Further studies on recreation in forests should investigate the lastingness of the effects of forest walks, applying a different methodology to verify this first result.

This study has several strengths, but also some weaknesses that need to be considered when interpreting its findings. As for the strengths, a citizen science approach shaped this study, as researchers and local people worked together to increase the non-timber values of the forest and to accompany this change with research for several years. This long-standing collaboration enabled a transfer of scientific knowledge into practice and vice versa. The study design, the involvement of the local population in data collection and in data management, before and after furnishing, are strengths of the study. We made a great effort to investigate the impact of furnishing using different approaches. Effects on the places and on the psychological well-being of the visitors, as well as assessments of the furniture, were accompanied by questions on the forest visit. Our sample consisted of experienced and potential future forest visitors from the population covering different age groups. The composition in the study population is a further strength of the study.

While the study was carefully planned, some limitations were apparent. Due to the field-experimental design of the study, not all external influences, such as weather conditions, could be controlled. However, the sample size compensated for systematic biases. The post-furniture sample contained a higher proportion of females and participants who were visiting the forest for the first time. We controlled statistically for these differences. A study showed a strong impact of the season on the perceived quality of the landscape in commercial forests [25]. Further studies should investigate the places in the furnished forest in different seasons. We could not quantify the influence of the guides on the recorded effects. However, we controlled for potential impacts by having all guides trained to always

give the same instructions to the visitors and having multiple guides of different genders and expertise guiding the groups through the forest.

Educational boards are common in forests. Some studies have examined the influence of educational boards in the context of forest visits [27–29]. They reported also about recovery effects. Therefore, it would be interesting to investigate the combined effect of furnishing and educational boards on human restoration. Another interesting topic for future research would be to study guided versus unguided forest walks. This could contribute to the well-being of some trail users and to understanding the social dimension of forest visits [12]. The results of this study could be used to promote public health and well-being in the region, and to promote health tourism. Therefore, they also serve as a basis for regional development, diversification in forestry, and sustainable forest management. Stress reduction, health promotion, and prevention of lifestyle diseases are great challenges in coping with actual manifold crises these days. Therefore, visiting forests should be recommended for preserving people's good health. Through the citizen science approach, many different people visited the forest. Visitors became very familiar with the forest by participating in the research. The places in the forest are open to the public with or without attending guided tours. The forest can be visited alone, with a partner or in groups. We proved the beneficial effects of unfurnished forests as well as of furnished ones. Therefore, we conclude that furnishing in forests is not necessary in order to achieve the forests' beneficial effects on health and well-being of visitors.

For attracting visitors, however, furnishing seems important. The majority of forests in Austria are unfurnished and open for visitors. Unfurnished forests represent the usual condition. Furnishing a forest, therefore, can make this forest distinctive and unique. Professionals could use the furnished forest for educational or healthcare services. This could increase both the care for the ecosystem and the socio-economic value of the forest. In the case of health-promoting interventions, an appropriate place that supports the desired effect of an intervention could be visited. For the development of stress-relieving interventions, however, we refer to recent findings [50]. Cognitively challenging tasks may result in unintended effects. Cognitive effort could reduce the restoration of attention through mindfulness interventions. Mindfulness, and other engagement interventions in forests, should therefore be designed with appropriate care, comparable to what we recommend for furniture, to alleviate mental stress.

Sustainable forest management should enhance the timber and non-timber values of forests. However, great care must be taken in furnishing, in order to keep the forest's beneficial effects. We therefore recommend soft forest furnishing when aiming to foster psychological restoration while experiencing the natural features of the forest. Sustainable forest management for health seems important for health promotion of locals and tourists, as well as for the implementation of healing forests, used in rehabilitation. In all cases, soft forest furnishing may increase the socio-economic functions of forests. In times of climate change, great care must be taken to keep the forest's inherent quality, both as an intact ecosystem and accordingly as a resource for public health.

## 5. Conclusions

In this study, we examined the impact of furnishing a recreational forest, utilizing three different approaches through citizen science. Our findings showed that the furnishing had varying impacts on the four sites within the forest, with only the site that was rated the highest prior to furnishing benefiting from the changes. While the furniture received positive ratings from visitors, it did not significantly impact psychological well-being or connectedness during the guided forest visit. We found that the furniture caused higher levels of disturbance in sites with lower recreational qualities.

Overall, the study highlights the importance of matching the furniture to the natural features of the site to achieve optimal restorative effects. The positive feedback from visitors suggests that furnishing can make the forest more attractive for tourism, pedagogy, and health interventions. Therefore, the idea of soft forest furnishing seems applicable to other

forests that support health and well-being, particularly healing forests, and should be considered in sustainable forest management.

**Author Contributions:** Conceptualization, R.C. and M.S.S.; data curation, R.C. and M.S.S.; formal analysis, R.C. and M.S.S.; funding acquisition, R.C. and D.H.; investigation, R.C. and M.S.S.; methodology, R.C. and M.S.S.; project administration, R.C.; resources, R.C. and D.H.; software, M.S.S.; supervision, R.C. and D.H.; validation, D.H.; visualization, R.C. and M.S.S.; writing—original draft, R.C., M.S.S. and D.H.; writing—review and editing, R.C., M.S.S. and D.H. All authors have read and agreed to the published version of the manuscript.

**Funding:** This research was partly funded by the Federal Ministry for Sustainability and Tourism of the Republic of Austria via a competitive grant of the University College for Agrarian and Environmental Pedagogy in 2018 and 2019 (funding to R.C. and M.S.S). Further funds were received from the Austrian Science Fund (FWF), Project Dr. FOREST, Grant number I 4411, which also partly funded the article processing charge (funding to D.H.).

**Institutional Review Board Statement:** The study was conducted in accordance with the Declaration of Helsinki. Ethical approval for this study was obtained from the Ethics Review Board of Upper Austria (07/2018).

**Informed Consent Statement:** Informed consent was obtained from all subjects involved in the study.

**Data Availability Statement:** The data presented in this study are available on request from the corresponding author.

**Acknowledgments:** We thank the local working group of the community of Adlwang for collaboration in the project, in particular Stefan Achathaler for his tireless commitment to the development of the project, and for providing images of the places in the Hallerwald. We further thank all officials, guides and participants who took part in the project, and Michael Kundi for methodological support.

**Conflicts of Interest:** The authors declare no conflict of interest.

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
