# Peer review of "Furnishing a Recreational Forest—Findings from the Hallerwald Case Study"

_forests, doi:10.3390/f14040836_

Round 1

Reviewer 1 Report

The first keyword can be written in lower case.

L. 51-58. forest infrastructure, includes educational boards. In a study by Korcz et al. 2021, it was indicated that educational boards may be a factor that does not raise vigor, i.e. may interfere with mental regeneration. Recommended for viewing - https://doi.org/10.3390/f12080993

Both the introduction and the discussion need a minor adjustment relative to more scientific publications. 

I suggest, for example, in the discussion to pay attention to the aspect of the guide during walks in the forest, since their presence can have a significant impact on the interpretation of nature, as well as the very well-being of trail users

https://hdl.handle.net/10520/EJC117161

https://doi.org/10.1093/jof/99.1.37

Author Response

Reviewer 1

Comment:

The first keyword can be written in lower case.

  1. 51-58. forest infrastructure, includes educational boards. In a study by Korcz et al. 2021, it was indicated that educational boards may be a factor that does not raise vigor, i.e. may interfere with mental regeneration. Recommended for viewing - https://doi.org/10.3390/f12080993

Response:

We thank you for the overall favorable evaluation of our manuscript. Your feedback and comments were very helpful to increase the quality of the content.

In response to your requests, we confirm that the first keyword in lower case, and we reviewed and incorporated the suggested publications.

Comment:

Both the introduction and the discussion need a minor adjustment relative to more scientific publications.

Response:

We agree and adjusted the text accordingly.

Amendment of Introduction:

“Some studies examined the influence of educational boards and of guides in the context of forest visits [27-29]. They reported also about recovery effects.”

Amendments of Discussion:

“Educational boards are common in forests. Some studies examined the influence of educational boards in the context of forest visits [27–29]. They reported also about recovery effects. Therefore, it would be interesting to investigate the combined effect of furnishing and educational boards on human restoration.”

“We could not quantify the influence of the guides on the recorded effects. However, we controlled for potential impacts by having all guides trained to always give the same instructions to the visitors and having multiple guides of different genders and expertise guiding the groups through the forest.”

Comment:

I suggest, for example, in the discussion to pay attention to the aspect of the guide during walks in the forest, since their presence can have a significant impact on the interpretation of nature, as well as the very well-being of trail users

https://hdl.handle.net/10520/EJC117161

https://doi.org/10.1093/jof/99.1.37

Response:

In response to this request, we adjusted the text accordingly.

Amendment to Discussion:

Another interesting topic for future research could be to study guided versus unguided forest walks. This could contribute to the well-being of some trail users and understanding of the social dimension of forest visits [12].

Again, we thank you or your valuable feedback!

Reviewer 2 Report

The paper has an interesting theme on influence of forest users' outcomes due to furnishing conditions. I would like to suggest followings to revise the manuscript.

1. The manuscript is need to expand stronger problem statement or justification, let say, why this study is needed in the Introduction.

2. In the Materials and Methods, the manuscript is needed to more specific explanation of the selection standards of four study areas. Why the study selected the study areas.

3. In the Study sample section, why the study selected criteria for voluntary  participants, especially, 3) immunization against tick-born encephalitis?

4. In the line 165, what "two data sets were excluded from analysis"?

5. The study used two different groups of participants, before and after furnishing. How the study argues the two samples have the same levels of dependent variables/outcomes such as restoration at the baseline? 

Author Response

Reviewer 2

Comment: The paper has an interesting theme on influence of forest users' outcomes due to furnishing conditions. I would like to suggest followings to revise the manuscript.

Response: We thank you for the overall favorable evaluation of our manuscript. Your feedback and comments were very helpful to increase the quality of the content.

Comment 1: The manuscript is need to expand stronger problem statement or justification, let say, why this study is needed in the Introduction.

Response 1: We agree and adjusted the text accordingly. In response to your requests, we made the following amendment to the Introduction:

“In the last few years, the interest in visiting recreational forests and the establishment of healing forests has been increasing [10]. This is accompanied by efforts to furnish such forests. Therefore, it is important to investigate the influence of furnishing on the recreational function of forests and to make recommendations on design.”

Comment 2: In the Materials and Methods, the manuscript is needed to more specific explanation of the selection standards of four study areas. Why the study selected the study areas.

Response 2: In response to your requests, we made the following amendment to the Materials and Methods section:

“Forest owners chose four places to be studied. Sites should represent different stand types; show various characteristics of the forest. Further, they should be accessible by different user groups, such as people in different phases of life or those with health and mobility restrictions. Therefore, the four places under study are all located near a well-kept forest path and are easily accessible for visitors.”

Comment 3: In the Study sample section, why the study selected criteria for voluntary participants, especially, 3) immunization against tick-born encephalitis?

Response 3: We agree and adjusted the text accordingly. Amendment to study Sample section:

“Additionally, 4) immunization against tick-born encephalitis was required, as it is a public health threat in Austria, and we did not want to put participants in danger of getting sick.”

Note that we only changed the order of the criteria to improve reading flow after amendment.

Comment 4: In the line 165, what "two data sets were excluded from analysis"?

Response 4: To increase clarity, we deleted this statement, as it was just a statistical note not necessary for understanding the content.

Comment 5: The study used two different groups of participants, before and after furnishing. How the study argues the two samples have the same levels of dependent variables/outcomes such as restoration at the baseline?

Response 5: We are thankful for this comment. We are aware that baseline values between groups could indeed appear as a statistical problem. Therefore, we checked for this issue. By entering baseline values as a predictor in the regression model we controlled for differences in baseline values between groups. The interaction term representing the effect of furnishing was non-significant in all models – validating the results of the t-tests. We hope that this explanation is ok for you.

Again, we thank you or your valuable feedback!

Round 2

Reviewer 2 Report

The manuscript has revised based on my reviewing comments.